# Diethanolamine Modified Perovskite-Substrate Interface for Realizing Efficient ESL-Free PSCs

**DOI:** 10.3390/nano13020250

**Published:** 2023-01-06

**Authors:** Sajid Sajid, Salem Alzahmi, Dong Wei, Imen Ben Salem, Jongee Park, Ihab M. Obaidat

**Affiliations:** 1Department of Chemical & Petroleum Engineering, United Arab Emirates University, Al Ain P.O. Box 15551, United Arab Emirates; 2National Water and Energy Center, United Arab Emirates University, Al Ain P.O. Box 15551, United Arab Emirates; 3College of Physics and Energy, Fujian Normal University, Fuzhou 350007, China; 4College of Natural and Health Sciences, Zayed University, Abu Dhabi P.O. Box 144534, United Arab Emirates; 5Department of Metallurgical and Materials Engineering, Atilim University, Ankara 06836, Turkey; 6Department of Physics, United Arab Emirates University, Al Ain P.O. Box 15551, United Arab Emirates

**Keywords:** ESL-free perovskite solar cell, diethanolamine, interface modification, high performance

## Abstract

Simplifying device layout, particularly avoiding the complex fabrication steps and multiple high-temperature treatment requirements for electron-selective layers (ESLs) have made ESL-free perovskite solar cells (PSCs) attractive. However, the poor perovskite/substrate interface and inadequate quality of solution-processed perovskite thin films induce inefficient interfacial-charge extraction, limiting the power conversion efficiency (PCEs) of ESL-free PSCs. A highly compact and homogenous perovskite thin film with large grains was formed here by inserting an interfacial monolayer of diethanolamine (DEA) molecules between the perovskite and ITO substrate. In addition, the DEA created a favorable dipole layer at the interface of perovskite and ITO substrate by molecular adsorption, which suppressed charge recombination. Comparatively, PSCs based on DEA-treated ITO substrates delivered PCEs of up to 20.77%, one of the highest among ESL-free PSCs. Additionally, this technique successfully elongates the lifespan of ESL-free PSCs as 80% of the initial PCE was maintained after 550 h under AM 1.5 G irradiation at ambient temperature.

## 1. Introduction

Within the last few years, PSCs have progressed significantly in their PCEs, rising from 3.8 to 25.7% [1]. PSCs are conventionally fabricated using two architectures: planar and mesoscopic [2,3,4,5,6]. Mesoscopic PSCs are generally made up of an electron-selective layer (ESL), a mesoscopic TiO_2_ or Al_2_O_3_ scaffold, a perovskite layer, a hole-selective layer (HSL), and back and front electrodes [7]. Further research found that PSCs can produce a PCE of over 15% utilizing a straightforward planar architecture, achieving high efficiency without the need for a mesoscopic scaffold [8,9,10,11,12,13,14,15,16,17]. Due to perovskite’s ambipolar characteristic, additional device architectural simplification is possible in order to fabricate efficient PSCs without using complicated fabrication methods [18,19]. In this scenario, HSL-free PSCs can be produced without manufacturing or utilizing costly organic HSLs [20,21].

The subsequent focus in PSCs is on the simplified fabrication technique that offers simple and inexpensive mass production in order to become economically relevant in the context of high efficiency and reasonable stabilities [22]. In this context, numerous researchers constructed ESL-free PSCs with reasonable PCEs [23]. ESL-free PSCs have a high device stability potential in addition to decent PCEs and do not need the time-consuming manufacturing of ESLs, particularly the numerous high-temperature processes required to create TiO_2_ ESL. Since it is known that TiO_2_ causes the perovskite layer to decompose by creating oxygen vacancies, the removal of the TiO_2_ ESL is mostly responsible for the high device stability [24]. However, because of the inefficient electron transport and the associated carrier recombination at the interface of transparent conductive oxides (TCO) and perovskite, the performance of ESL-free PSCs is still inferior to that of conventional devices [25,26,27]. An additional reason for the low device efficiency of the ESL-free PSCs is the rough and uneven perovskite film when no ESL is fabricated, which will produce many open grain boundaries and lead to severe recombination as a result of the shunt pathways between the HSL and TCO. Surface modification methods were recently suggested to reduce charge-carrier recombination at the TCO/perovskite interface and to initiate charge collection at the relevant contact [28,29].

The efficacy of the resulting ESL-free PSCs is enhanced by carefully adjusting the compositions, architecture, and morphology of the perovskite layer as well as pre-treating the TCO (UV/O_3_ irradiation) to prevent carrier recombination in the perovskite or at the TCO/perovskite interface [24]. These ESL-free PSCs, however, displayed relatively lower PCEs (<18%) than those of their ESL-containing counterparts. Other methods for improving device performance and surface modification of TCO substrates include bathocuproine [25], polar non-conjugated small molecule modifier [30], and hydroxyethyl-functionalized imidazolium iodide ionic liquid [31], and self-assembled fullerene monolayer [32]. Recently, PCEs of ESL-free PSCs >19% have been reported employing sodium fluoride [33] and tetramethylammonium hydroxide [34] as TCO modifiers. Nevertheless, most of the modifications in the aforementioned approaches need additional or difficult processes, which consumes time and energy. These modifiers are also regarded for their hygroscopicity, insulating properties, and high material costs.

In the present study, we used a low-cost, very simple Indium-Tin Oxide (ITO)-surface modification with DEA to enhance the ITO/perovskite interaction and perovskite crystallization. Because amine groups typically interact with the ITO surface and Pb-based substances are reported to link with –OH groups, we chose DEA, which contains both –OH and amine groups, as the contact modifier between ITO and perovskite [35]. As anticipated, the DEA molecular interlayer interacted with ITO and perovskite, which strengthened the interface contact and elevated the crystallinity of the perovskite. A dipole layer that is well-orientated for electron extraction was also produced by molecular adsorption. The electron extraction rate was significantly improved as a result, and the ESL-free PSC with the ITO/DEA/Perovskite/spiro/Au device structure demonstrated significantly better performance (PCE: 20.77%) than an ITO/Perovskite/spiro/Au device (PCE: 18.65%).

## 2. Experimental Procedure

### 2.1. Solution Preparation

Perovskite solution was obtained by mixing anhydrous dimethylformamide/dimethylsulfoxide (600 mg/78 mg) with 0.15 M of formamidinium lead iodide (FAI), 0.85 M of methylammonium lead iodide (MAI), and 1.025 M of PbI_2_. This solution was stirred for 4 h in a glove box. Spiro-OMeTAD was dissolved in chlorobenzene at a concentration of 80 mg/mL to make the precursor for the hole-selective layer. Tert-butylpyridine (28.5 μL in 1 mL of chlorobenzene) and lithium bis-(trifluoromethanesulfonyl) imide (8.75 mg/mL) were then added as additives, and the mixture was stirred in a glove box for 6 h.

### 2.2. Device Fabrication

The ITO substrates were ultrasonically cleaned with detergent solution, deionized water, acetone, ethyl alcohol, and deionized water for 20 min, respectively. The ITO substrates were exposed to UV-ozone for 20 min after drying. The cleaned ITO substrates were spin-coated with the DEA pure solvent for 30 s at 2000 rpm and then dried for 10 min at 100 °C. The bare cleaned ITO and DEA-treated ITO substrates were spin-coated with the perovskite precursor at 4000 rpm for 25 s to form the perovskite layers. Eighteen seconds before the end of the spin-coating process, 0.8 mL of diethyl ether was dripped on the perovskite surface. The resulting perovskite layers were heated for 15 min at 130 °C before naturally cooling to ambient temperature. Subsequently, both perovskite layers were spin-coated with the Spiro-OMeTAD solution for 30 s at 4000 rpm. The devices were finished by evaporating 60 nm Au electrodes onto the ITO/perovskite/spiro-OMeTAD and ITO/DEA/perovskite/spiro-OMeTAD.

### 2.3. Characterizations

A Thermo Fisher Scientific (Shanghai, China) ESCALAB 250Xi under 10^−9^ Torr vacuum with a monochromatic Al-K_α_ X-ray source was used to collect the XPS spectra. A scanning electron microscope (Hitachi, Chiyoda, Japan, S-4800) and an atomic force microscope (Agilent Keysight (Beijing, China) AFM-5500) were utilized to study the morphologies. The perovskite’s crystallinity was examined using an X-ray diffractometer (Bruker D8 Advance (Bruker AXS Inc.), Cu-K_α_ radiation of 0.15406 nm). The UPS spectra were obtained using the ESCALAB 250Xi Contact angle images were obtained with an Ossila Contact Angle Goniometer. The FT-IR spectra were collected by using a Jasco (Tokyo, Japan) FT-IR spectrometer. Using a UV-Vis spectrophotometer (UV-2600), the absorption spectrum was measured. The steady-state PL spectra of the produced samples were looked at utilizing an Edinburgh PLS 980. Utilizing a transient state spectrophotometer (Edinburgh Institute F900) and a 485 nm laser, the TRPL decay of the perovskite layers was monitored. Under AM 1.5 G illumination with a power intensity of 100 mW cm^−2^, the *J*-*V* characteristic curves were measured with a source meter (Keithley (Cleveland, OH, USA) 2400) utilizing forward (−0.1 to 1.2 V) or reverse (1.2 to −0.1 V) scans from a solar simulator (XES-301S+EL-100). The delay time was set to 10 ms, and the step voltage was set to 12 mV. The EQE was calculated using QE-R systems (Enli Tech., Shanghai, China). The EIS measurement was carried out using an electrochemical workstation (Zahner Zennium). The EIS data were fitted with help of ZSim-software version 3.20 with equivalent circuit parameters of R_1_ = 0.116, C = 0.00001 and R_2_ = 54.1 (for ITO-based device), and R_1_ = 0.116, C = 0.000001 and R_2_ = 29.6 (for DEA-treated ITO-based device).

## 3. Results and Discussion

The cleaned ITO substrates were first spin-coated with DEA at 2000 rpm for 30 s, then dried at 100 °C for 10 min. X-ray photoelectron spectroscopy (XPS) was carried out to investigate the effect of DEA treatment on the surface chemistry of ITO. The XPS survey spectra of bare ITO and DEA-treated ITO are presented in Figure 1. All spectra were calibrated using the C 1s peak of 284.9 eV binding energy as a reference point. The bare ITO and DEA-treated ITO share the same peaks except at 400.4 eV, which is assigned to N 1s, showing that the DEA successfully adsorbed on the ITO substrate, as can be seen in Figure 1a,b. Figure 1c displays the high-resolution N 1s spectra of ITO/DEA, which can be divided into three peaks at 401.4, 400.4, and 399.7 eV. Sigma-state, including species containing N, can be attributed to the peaks at 401.4 eV and 400.4 eV. Additionally, it has been stated that an amine group correlates to the peak at 399.7 eV [36,37]. The shift in the binding energy, as observed in the Sn 3d (Figure 1d) signals, is indicative of a chemical interaction between DEA and the ITO substrate. This suggests a dipole interaction between ITO and DEA that is mediated by an electric field, which lowers the attractive force of the Sn nuclei and increases the Sn atom’s outer-shell electron density [38].

To investigate the impact of DEA coating on the surface morphology of ITO, scanning electron microscopy (SEM) was conducted on bare ITO and DEA-treated ITO. Figure 2a,b displays the surface SEM images of ITO and ITO/DEA, respectively. As can be seen, both surfaces are compact without pinholes. However, because the DEA layer is so thin, it is hard to see the DEA-coated ITO surface clearly. Appendix A shows that this thin layer of DEA had little influence on ITO transparency across a wide spectrum of light, indicating a great potential for high photocurrent output. The morphologies of the bare ITO and DEA-treated ITO were investigated with the help of atomic force microscopy (AFM), as shown in Figure 2c,d. The alteration in the morphology following DEA coating is not very noticeable, similar to the SEM images. However, in the DEA-treated ITO substrate, a reasonable decrease in the root-mean-square (RMS) of surface roughness from 2.8 to 0.9 nm is observed. The DEA coating may considerably minimize the surface defect states and promote the deposition of high-quality perovskite thin film because of the reduced roughness. We also used Kelvin probe force microscopy (KPFM) to examine bare and DEA-treated ITO, as displayed in Appendix A. This revealed that the surface potential distribution is consistent with the aforementioned surface morphologies, and the average potential of the ITO/DEA substrate is −110 mV compared to the 0.18 mV of bare ITO, which indicates desirable Fermi-level of the ITO/DEA and better band alignment with perovskite [24,39].

In order to get highly efficient PSCs in ESL-free architectures, one of the main obstacles is the energy-level mismatch between ITO and the perovskite, so it is essential to investigate how DEA affects the energy levels of ITO. Figure 3a,b illustrates the outcomes of an ultraviolet photoelectron spectroscopy (UPS) experiment. For the purpose of calculating *E_cutoff_* (secondary electron edges) and *E_onset_* (Fermi edges), the intercepts of the tangents of the peaks with the extrapolated baselines were used. By estimating the difference between the incident photon energy (light source He I, 21.22 eV) and *E_cutoff_*, the Fermi-level (*E_F_*) values of the bare and DEA-treated ITO were quantitatively determined. The UPS show an *E_cutoff_* of 16.64 eV and 17.21 eV for bare ITO and ITO/DEA, respectively. The E_F_ values of bare ITO and ITO/DEA were estimated to be −4.58 and −4.01 eV, respectively, using the following equation.
(1)$$E_{F}=E_{cutoff}-21.22\;\mathrm{eV}\;$$

The *E_F_* of the ITO was significantly shifted by −0.57 eV after DEA treatment, which would be advantageous for the cathode’s ability to collect electrons. This is because, following DEA treatment, a favorable dipole moment formed on the surface of the ITO. As stated in previous studies, Pb-based compounds interact with –OH groups, while the amine groups in DEA tend to link with the ITO surface [34,35]. As can be seen, the DEA modification resulted in a −0.57 eV shift in the Fermi level of ITO, which should be attributed to a chemical reaction between DEA and ITO, as supported by the FT-IR data (Appendix A). The appearance of the N-atom signal in Figure 1 demonstrates the interaction between the N-atom of DEA and ITO. The -NH- (from the amine group) should function as an electron donor to engage in chemisorption and modify the energy levels of ITO. To examine the surface wettability of ITO treated with DEA, the water contact angle was measured (Figure 3c). Since DEA possesses hydrophilic –OH and amine groups, the water contact angles of the ITO surface decrease from 70.1° to 59.6°, showing greater wetting capabilities of DEA-treated ITO. The superior wettability can decrease the nucleation barrier and aid in the formation of a high-quality perovskite layer.

The surface morphologies and cross-sectional images of the perovskite layers prepared on bare ITO and ITO/DEA substrates were taken with help of SEM, as depicted in Figure 4. It is evident that the perovskite layers that were formed on ITO/DEA have densely packed, considerably larger grains. In order to facilitate charge transfer, the perovskite layer formed on ITO/DEA substrate displays entire coverage without any grain boundary gaps. The larger perovskite grains and improved morphology on the ITO/DEA substrate can be attributed to the favorable interaction of the –OH group with Pb^2+^ [35], which promotes the formation of larger grains and passivation of the perovskite’s grain boundaries. A schematic representation of the DEA interaction is shown in Appendix A, where the –OH groups of the DEA would interact with the Pb-atoms in the perovskite thin film while the N-atom of DEA tends to create a chemical connection with the Sn atom on the ITO surface. According to other researchers, the interaction between the –OH group and Pb^2+^ can improve the penetration of PbI_2_ into a TiO_2_ mesoporous layer [40]. Through the direct mixing of DEA solution and PbI_2_ powder, we were able to further test the possible interaction between DEA and PbI_2_. Intriguingly, adding PbI_2_ to DEA has significantly altered the appearance, likely creating new products. As seen in Appendix A, the colorless DEA solution and the yellow PbI_2_ powder were combined to create an entirely distinct product that was white in color. The white mixture turns into a colorless liquid after heating (Appendix A). To further confirm these interactions, Fourier-transform infrared (FT-IR) spectra in the wavenumber range of 600 to 4000 cm^−1^ were carried out, as shown in Appendix A. The spectrum of the ITO substrate shows typical broad bands below 800 cm^−1^, which is assigned to the stretching vibration of In-O or Sn-O bonds [41]. The DEA-modified ITO shows broad and strong bands below 800 cm^−1^. In addition to the broad and strong bands, the DEA-modified ITO displays an additional sharp and strong peak at 630 cm^−1^, which is ascribable to the stretching vibration of the Sn-N bond. Therefore, it is reasonable to assume that the DEA and ITO substrate were linked by a Sn-N bond, exposing the –OH groups for interaction with the perovskite layer that was later added. The DEA and PbI_2_/DEA were characterized using FT-IR spectroscopy, and the variation in –OH absorption peak was monitored. Broad absorption peaks can be seen in the DEA and DEA/PbI_2_ spectra between 1052 and 1458 cm^−1^, which corresponds to the scissoring vibration of the –OH group, as seen in Appendix A. The peak locations altered in the PbI_2_/DEA sample from 1052–1458 cm^−1^ to 1044–1448 cm^−1^, indicating a connection between PbI_2_ and the –OH groups of DEA. As seen in Appendix A, the addition of PbI_2_ to DEA has significantly altered the appearance, perhaps creating new compounds. These observations point to a significant interaction between DEA and PbI_2_. In perovskite deposition, the exposed –OH groups on the ITO surface would interact favorably with the PbI_2_, which is present in the perovskite precursor solution. An even and complete coverage of the perovskite layer would benefit from this advantageous interaction, ensuring good interface contact with the ITO/DEA substrate.

The formation of high-quality perovskite layers may improve the performance of corresponding PSCs by facilitating charge transport and reducing recombination losses [42]. In addition to morphological characteristics, X-ray diffraction (XRD) and UV-vis absorption spectra were used to study how DEA deposition affected the structure and optical performance of the perovskite layers. The high-intensity peaks from the XRD pattern of the perovskite layer formed on ITO/DEA displayed slightly greater crystallinity compared to bare ITO, as shown in Figure 5a. Additionally, the perovskite layer formed on the ITO/DEA substrate exhibits slightly greater absorption capabilities throughout the entire spectral region as evidenced by UV-vis absorbance spectra in Figure 5b. Steady-state photoluminescence (PL) and time-resolved PL (TRPL) measurements were used to analyze the charge-carrier dynamics of perovskite layers grown on glass, bare ITO, and ITO/DEA substrates, as shown in Figure 5c,d. When compared to perovskite layers formed on glass and glass/ITO, the PL spectra in Figure 5c show a considerable drop in intensity for the ITO/DEA substrate, suggesting higher electron transfer to the corresponding layer [43] and lower recombination of charges [44]. A significant quenching effect for the perovskite layer deposited on the ITO/DEA substrate can be seen in the TRPL spectra given in Figure 5d, demonstrating the improved electron extraction capability from the perovskite layer to the DEA-treated substrate. The results were fitted by a biexponential decay function with the following equation.
(2)Y=k1e−xτ1+k2e−xτ2
where *τ*_1_ and *τ*_2_ represent decay factors that are closely related to non-radiative recombination by defects and a component of bulk recombination, respectively. Appendix A shows decreased values for *τ*_1_ and *τ*_2_ for the perovskite layer deposited on the ITO/DEA substrate, indicating minimized recombination losses.

We fabricated PSCs with and without DEA for the evaluation of photovoltaic parameters after carefully examining the morphology and substantial role of the DEA at the ITO/perovskite interface. The schematic diagram of the as-prepared device and energy levels of each layer are depicted in Appendix A. The cross-sectional SEM images, current-voltage (*J*-*V*) characteristic curves, and photovoltaic parameters of the as-prepared PSCs are depicted in Figure 6 and Table 1, respectively. The PSC employing bare ITO displays an open-circuit voltage (*V_oc_*) of 1.096 V, a short-circuit current density (*J_sc_*) of 22.94 mA cm^−2^, a fill factor (FF) of 74.19%, and a power conversion efficiency (PCE) of 18.65% under reverse scan condition. In comparison, the DEA-modified PSC shows improved performance of up to PCE of 20.77%, a *V_oc_* of 1.109 V, a *J_sc_* of 24.45 mA cm^−2^, and an FF of 76.60% under the same scan condition. It is noteworthy that the DEA-modified PSC exhibits significantly better performance and less hysteresis in the *J*-*V* curves. The better layer quality and improved charge-carrier transport at the ITO/DEA/perovskite interface might be primarily responsible for the improved performance and decreased hysteresis, which is consistent with the dark *J*-*V* curves, as discussed below, where we observed a decrease in the density of trap states in the DEA-modified PSCs.

To further confirm the charge-carrier dynamics and charge-carrier recombination in the devices with bare ITO and ITO/DEA, the dark *J*-*V* measurement was carried out. As seen in Figure 7a, the device with DEA has a lower current density than the device with bare ITO. This shows that the DEA/perovskite interface has efficient charge transport. In order to quantitatively assess the trap density in the as-prepared devices, we also fabricated electron-only devices (ITO/Perovskite/PCBM/Au and ITO/DEA/perovskite/PCBM/Au), as shown in Figure 7b. The trap density (*N_trap_*) can be determined with the equation
(3)$$N_{trap}=\frac{2\varepsilon_{0}\varepsilon V_{TFL}}{eL^{2}}\;$$
where $\varepsilon_{0}$, ε, *V_TFL_*, *e*, and *L* are the vacuum permittivity, relative dielectric constant, trap-field limit voltage, elementary charge, and perovskite layer thickness, respectively. Since *N_trap_* and *V_TFL_* are linearly related, a device with a smaller *V_TFL_* will exhibit fewer defects overall. As seen in Figure 7b, the device with DEA exhibits a significantly lower *V_TFL_* (0.25 V) than the device with bare ITO (0.33 V), indicating DEA successfully reduces the defect density. The smooth morphology with fewer grain boundaries and the high-quality crystallinity of the perovskite layer deposited on the ITO/DEA substrate are responsible for the reduced trap density. Electrochemical impedance spectroscopy (EIS) measurements were conducted in order to learn more about how DEA affects the interfacial charge transport and recombination processes in the PSCs. Figure 7c displays the Nyquist plots of the corresponding devices with a bias voltage of 0.9 V under dark conditions and a frequency range of 1 Hz to 1 MHz. At higher frequencies, the semicircle stands in for the charge transport resistance (*R_ct_*), which originates from the substrate and perovskite contact. Because ITO/DEA made better contact with the perovskite surface, the DEA-treated PSC showed a lower *R_ct_* value (29.76 kΩ) than the ITO-based device (*R_ct_* of 54.06 kΩ), indicating more efficient electron transfer and electron extraction at the corresponding interface. Additionally, measurements of external quantum efficiency (EQE) were made in order to assess how well the *J*-*V* and EQE curves agreed. The matching *J_sc_* values integrated from the EQE curves (Figure 7d) for both devices are consistent with *J*-*V* characteristic curves. Here, we noticed a minor discrepancy between the EQEs of the PSCs and the absorption spectra of the perovskite layers prepared on ITO and DEA-treated ITO substrates in the range of 400–500 nm. This can be attributed to the fact that the absorption spectra do not represent an external property of the PSCs but rather an internal property of the perovskites (originating from the distinctive energy-band structure) [45,46,47]. Moreover, the EQE spectrum encompasses all excitonic and/or thermal broadening effects, along with the architectural features of the PSC, and this reflects the occupied density of states as specified in the entire PSC rather than in individual perovskites.

Furthermore, we acquired performance data from 9 PSCs with bare ITO and ITO/DEA, respectively, to show the reproducibility of the devices. The ITO/DEA-based PSCs exhibit improved performance and reproducibility when compared to PSCs with bare ITO, as shown in Appendix A. In addition to the high PCEs, the PSCs should have long-term stability. Figure 8a illustrates the tracking of the steady-state PCEs of the PSCs with ITO/DEA and bare ITO under AM 1.5 G at maximum power at 0.9 V bias voltage and ambient temperature. After 120 s of continuous illumination, the ITO/DEA-based device showed a stabilized PCE of 19.71%, which is considerably near the highest PCE of 20.77% for the same PSC. Additionally, for 550 h of AM 1.5 G irradiation, the stabilities of the unsealed PSCs made with bare ITO and ITO/DEA substrates were evaluated every 34 h. After each stability test, the tested PSCs were kept in a glass oven at 60% humidity and 25 °C. Figure 8b demonstrates how the stability of the ITO/DEA-based PSCs exceeded that of the ITO-based devices. After 550 h, the ITO/DEA-based PSC still had 80% of the original PCE. The bare ITO-based PSCs under the same testing conditions only retained 37% of the initial PCE, demonstrating poor stability and quick degradation. The degradation of ITO-based PSCs is associated with rapid erosion because the perovskite layer formed on the bare ITO substrate has a defective surface morphology. As mentioned above, high-quality perovskite layers were created on the DEA-treated ITO substrates, with compact, large grains, and reduced grain boundaries. Starting from the grain boundaries makes it considerably easier to decompose perovskite films. Less oxygen, water, etc. will intermix within the grain boundaries of the perovskite films because there are fewer grain boundaries. The stability of PSCs made using DEA-treated ITO substrates is ultimately improved by the perovskite film with fewer grain boundaries, which effectively shields perovskite films from water penetration. Conversely, erosion occurs more rapidly in the perovskite thin films generated using the ITO substrates because these have a considerably rougher surface with smaller grain sizes and numerous grain boundaries.

## 4. Conclusions

We used DEA to generate Sn-N and –OH–Pb chemical linkages between the perovskite layer and the ITO substrate. The interface contact, perovskite crystallinity, and film quality have all increased due to these chemical interactions, and the perovskite layer’s grain boundaries and trap states have decreased. Based on the creation of interfacial dipole layers, our results imply that the addition of DEA can modify the band misfit at the ITO/perovskite contact, hence reducing the voltage deficits. Transient PL, TRPL, and IS measurements revealed that the rates of electron extraction and charge transfer have dramatically improved. The PCE of the PSCs has significantly increased as a result, rising from about 18.65% to 20.77%. Additionally, DEA modification has increased the stability of the PSCs, with 80% PCE retention even after 550 h without encapsulation. This is likely because the DEA modification improved the interfacial contact and thin-film quality of the perovskite. Our research opens up scientific pathways for the facile fabrication of ESL-free PSCs.

## Figures and Tables

**Figure 1 nanomaterials-13-00250-f001:**
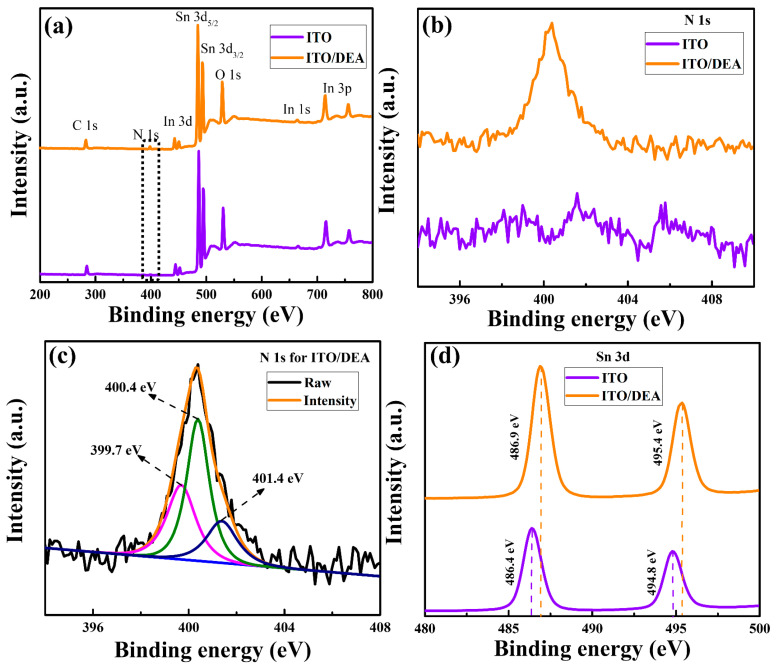
XPS spectra of ITO substrate and ITO/DEA substrate (**a**), N 1s spectra of ITO substrate and ITO/DEA substrate (**b**), analysis of N 1s spectra of ITO/DEA substrate (**c**), and Sn 3d spectra of ITO substrate and ITO/DEA substrate (**d**).

**Figure 2 nanomaterials-13-00250-f002:**
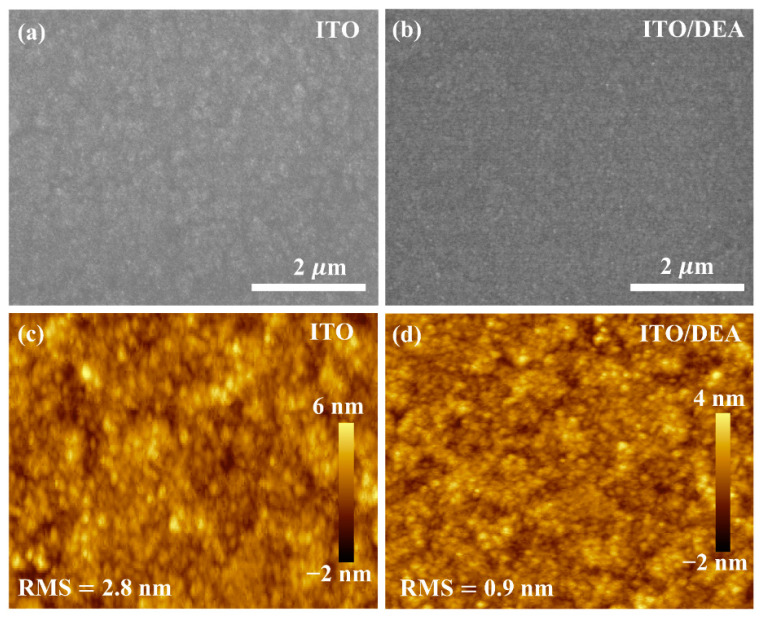
SEM images of the ITO substrate (**a**), ITO/DEA substrate (**b**), AFM images of ITO substrate (**c**), and ITO/DEA substrate (**d**).

**Figure 3 nanomaterials-13-00250-f003:**
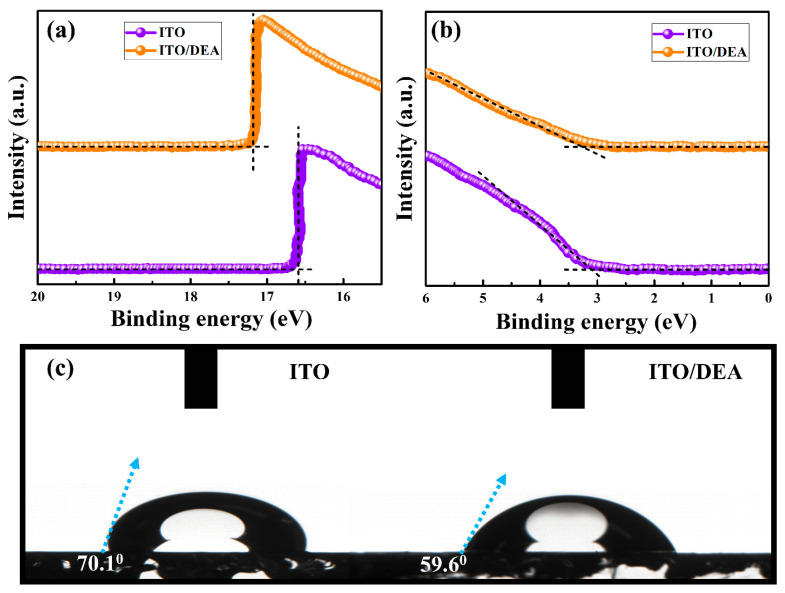
UPS spectra of ITO substrate and ITO/DEA substrate (**a**,**b**), and the contact angle photographs of water droplets on ITO substrate and ITO/DEA substrate (**c**).

**Figure 4 nanomaterials-13-00250-f004:**
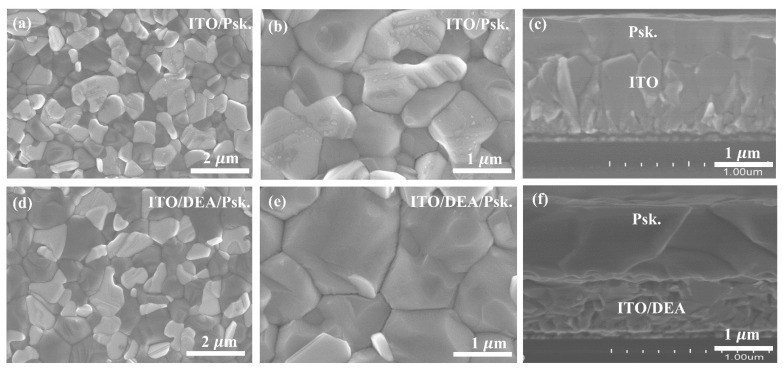
SEM images of the surface and cross-section of the perovskite (Psk.) layers prepared on ITO substrates (**a**–**c**) and ITO/DEA substrates (**d**–**f**), respectively.

**Figure 5 nanomaterials-13-00250-f005:**
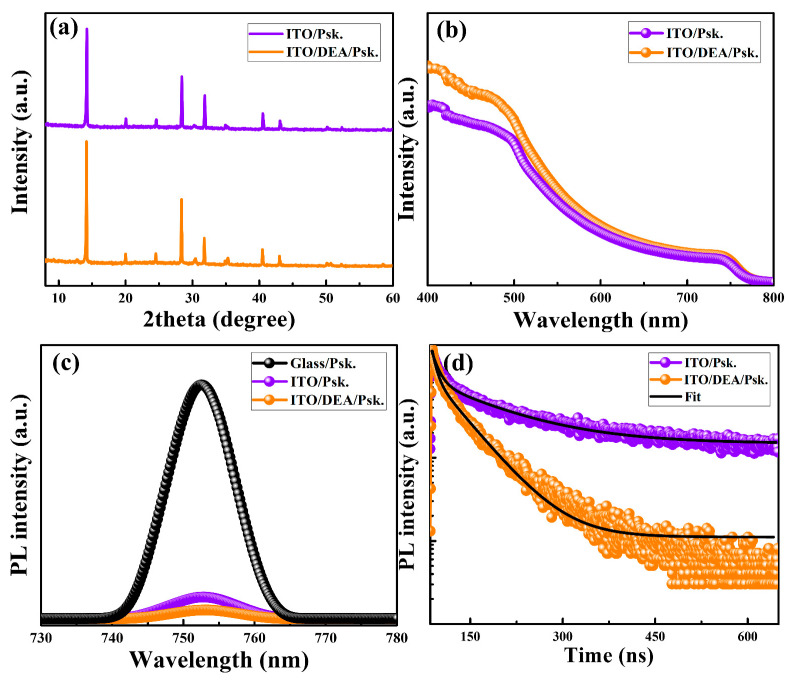
XRD patterns (**a**), UV-Vis spectra (**b**), PL-spectra (**c**), and TRPL-spectra (**d**) of the ITO/perovskite and ITO/DEA/perovskite, respectively.

**Figure 6 nanomaterials-13-00250-f006:**
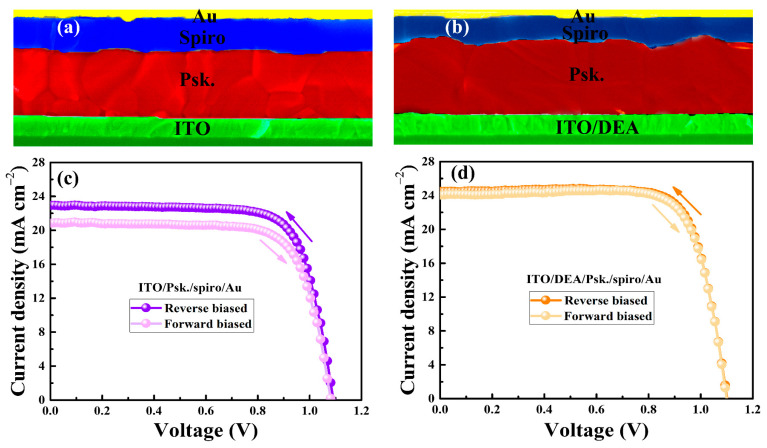
Cross-sectional SEM images of as-fabricated devices with ITO substrate (**a**) and ITO/DEA substrate (**b**). The *J*-*V* characteristic curves of devices fabricated with ITO substrate (**c**) and ITO/DEA substrate (**d**).

**Figure 7 nanomaterials-13-00250-f007:**
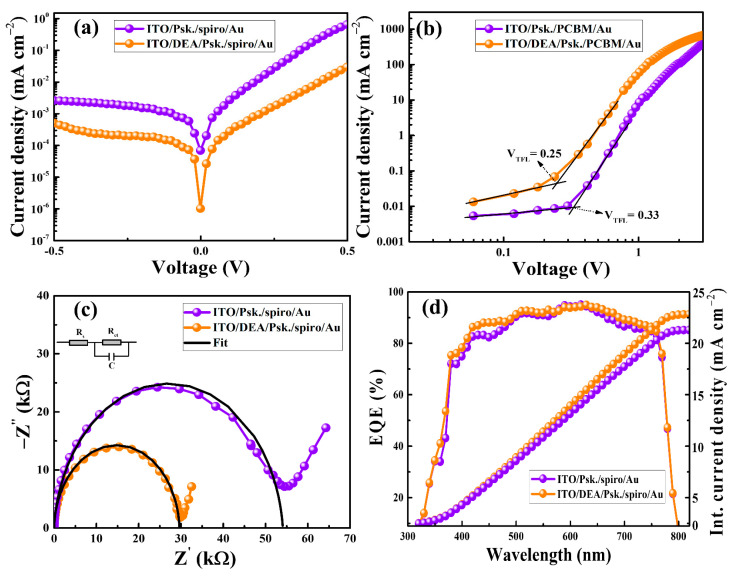
Dark *J*-*V* curves of as-prepared devices (**a**) and electron-only devices (**b**), EIS measurement (**c**), and EQEs with integrated *J_sc_* of the PSCs prepared using ITO substrates and ITO/DEA substrates (**d**).

**Figure 8 nanomaterials-13-00250-f008:**
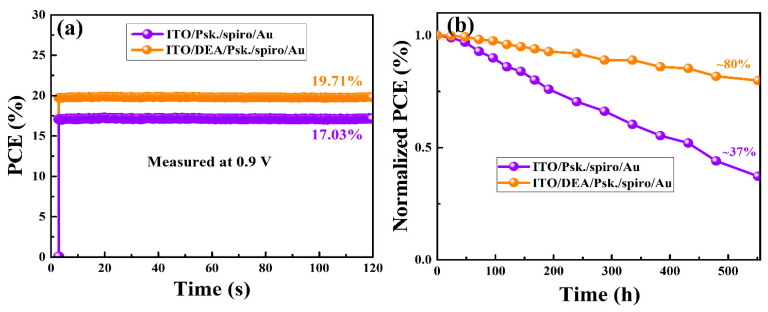
Stability analyses of the PSCs prepared with ITO substrates and ITO/DEA substrates. PCE at maximum power point tracking under continuous illumination of AM 1.5 G irradiation (**a**) and Normalized PCE for 550 h under AM 1.5 G irradiation (**b**).

**Table 1 nanomaterials-13-00250-t001:** Photovoltaic parameters of the PSCs under forward bias and reverse bias.

Device	Scan	*V_oc_* (V)	*J_sc_* (mA cm^−2^)	FF (%)	PCE (%)
ITO/Psk./spiro/Au	Reverse	1.096	22.94	74.19	18.65
	Forward	1.083	20.83	74.24	16.75
ITO/DEA/Psk./spiro/Au	Reverse	1.109	24.45	76.60	20.77
	Forward	1.096	24.02	77.94	20.52

## Data Availability

All the data presented in the manuscript can be obtained from the corresponding authors by reasonable request.

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
