# Peer review of "Diethanolamine Modified Perovskite-Substrate Interface for Realizing Efficient ESL-Free PSCs"

_nanomaterials, 2023, doi:10.3390/nano13020250_

Round 1

Reviewer 1 Report

The manuscript „Diethanolamine modified perovskite-substrate interface for realizing efficient ESLs-free PSCs“ by Sajid Sajid, Salem Alzahmi, Dong Wei, Imen Ben Salem, Jongee Park, Ihab M. Obaidat investigates the use of diethanolamine (DEA) to modify ITO as electron selective contact. The results are interesting and could be important for the scale up of perovskite photovoltaics. However, there are still some points that the authors need to satisfactorily address before publication may proceed:

1.    The authors should consider to add a schematic showing the different energy levels of the relevant layers of the solar cells.

2.    On page 6, the authors mention that new compounds might have been created by adding PbI2 to DEA. Have the authors considered to isolate these compounds for further investigation?

3.    On page 6, lines 210 – 213, the authors mention the use of TRPL to investigate the recombination dynamics. However, why should this type of biexponential describe non-radiative recombination and bulk recombination? Assuming non-radiative recombination is equivalent to band-to-band recombination, then this would follow a relationship of dn/dt = k_2*n^2 whereas bulk or Shockley-Read-Hall recombination would follow dn/dt = k_1*n (see also DOI: 10.1016/j.progpolymsci.2013.08.008). This is not represented by the biexponential shown in the manuscript and thus the conclusions regarding recombination are questionable. The authors should revisit this specific part of their manuscript.

4.    The authors should label the layers of their SEM images with what they suspect these layers to consist of.

5.    On page 8, line 243, the authors claim that the DEA/Perovskite interface has efficient charge transport just based on a reduced dark JV-curve. Can this conclusion be drawn just from the dark JV-curve? Usually more sophisticated measurements, such charge extraction methods or correlating JV-curves to charge carrier densities are necessary for this. The authors should either consider adding these measurements, or clarify their statement.

6.    In general, equation should be put in their own, separate line and numbered for better readability.

7.    Regarding the dark JV-curves, the authors should consider to determine the series and shunt resistance based on the differential resistance approach (see DOI: 10.1002/aenm.201901438; specifically Fig. 2). Also, they can cross-correlate these results to their EIS measurements.

8.    Regarding the EIS measurements, the authors need to list all components of the equivalent circuit model and their values to obtain the fit. The authors should then also make use of these determined value during their discussion (see for further use of EIS in solar cells DOI: 10.1063/5.0094955; specifically section B, Fig. 5).

9.    Minor comments regarding typos, grammar, and layout can be found in the commented PDF.

Reviewer 2 Report

The manuscript investigates ESL-free perovskite solar cells (PSCs) with diethanolamine (DEA) molecules between the perovskite and the ITO substrate. DEA suppressed charge recombination and delivered PSCs up to 20.77%.

The manuscript is well-organized with good-quality figures and logical scientific discussions. I believe that the manuscript meets the standards of Nanomaterials and will be well-appreciated by the science community. Thus, I recommend the manuscript to be accepted and published in Nanomaterials.

Reviewer 3 Report

In this work, authors used DEA as an interface modifier between TCO and perovskite to produce pinhole free devices with higher performance. The role of DEA is to form chemical bonds with both TCO and perovskite and improves TCO electrical properties, leading to enhanced charge transfer from perovskite layer to TCO electrode. The approach is simple but interesting. Below some comments about the manuscript are listed, which should be addressed prior to publication.

1)      Page 4 line 134-141, the authors mentioned that the DEA coting on ITO reduced the Fermi level of ITO by -0.57 eV. It is known that the Fermi level of material can be tuned by temperature or chemical doping, where the probability of occupation of energy levels changes by enhancement of kinetic energy or electrons of dopant into the electronic states of ITO. Could Authors please explain how the DEA thin film contributes to the change of Fermi level?

2)      Page 5-6, line 178-183- The FT-IR spectra, demonstrate an interaction between PbI2 and–OH group of DEA. For better monitoring the interaction of DEA with perovskite precursors, H-NMR need to be performed to observe possible bonding between –OH of DEA and PbI2. Or XPS study of ITO/DEZ/Psk could be used to confirm the impact of chemical bonding on Pb and O elements to support the proposed mechanism.

3)      Figure 4 c and f (FESEM cross section images) seem not correct. Based on scale bar, the thickness of perovskite layer on ITO/Psk is less than ITO (~750 nm), but on ITO/DEA/Psk the perovskite thickness in 1.5 times thicker than on ITO (~ 1500 nm). Could authors explain the reason of obtaining such thicker perovskite layer? Is that because of the effect of thin DEA film under perovskite layer? Also the scaling in images should be re-checked.

4)      Figure 5b, The ITO/DEA/Psk absorption has highest intensity than ITO/Psk at the range of 400-500 nm. After 500 nm the absorption difference of ITO/DEA/Psk gets smaller, but still higher. As the higher absorption should lead to higher photo-to-electron conversion, but such trend cannot be observed in EQE% (Figure 7d). The device should show a higher conversion rate at 400 nm, and then get closer to reference cell by 500 nm and in the rest of wavelength remain slightly higher than reference device. The data need to be re-checked.

5)      In Figure 8, the caption should be given to both (a) and (b) parts.

6)  In figure 8b, in the long term stability test, authors mentioned that ITO/DEA-based PSC had improved aging stability (43% better than ITO based device). Could the author explain how the DEA thin film deposited under perovskite film could suppress the degradation in perovskite phase while the device was stored under 60% humidity at RT?

Round 2

Reviewer 1 Report

The manuscript „Diethanolamine modified perovskite-substrate interface for realizing efficient ESLs-free PSCs“ by Sajid Sajid, Salem Alzahmi, Dong Wei, Imen Ben Salem, Jongee Park, Ihab M. Obaidat has improved during the revision process.

Reviewer 3 Report

The questions raised by the reviewer were properly answered.